# Production of Human Pluripotent Stem Cell-Derived Hepatic Cell Lineages and Liver Organoids: Current Status and Potential Applications

**DOI:** 10.3390/bioengineering7020036

**Published:** 2020-04-09

**Authors:** João P. Cotovio, Tiago G. Fernandes

**Affiliations:** iBB-Institute for Bioengineering and Biosciences and Department of Bioengineering, Instituto Superior Técnico, Universidade de Lisboa, Av. Rovisco Pais, 1049-001 Lisbon, Portugal; joaocotovio@tecnico.ulisboa.pt

**Keywords:** human pluripotent stem cells, hepatic cell lineages, hepatocyte differentiation, non-parenchymal liver cells, liver organoids, disease modeling, drug screening

## Abstract

Liver disease is one of the leading causes of death worldwide, leading to the death of approximately 2 million people per year. Current therapies include orthotopic liver transplantation, however, donor organ shortage remains a great challenge. In addition, the development of novel therapeutics has been limited due to the lack of in vitro models that mimic in vivo liver physiology. Accordingly, hepatic cell lineages derived from human pluripotent stem cells (hPSCs) represent a promising cell source for liver cell therapy, disease modelling, and drug discovery. Moreover, the development of new culture systems bringing together the multiple liver-specific hepatic cell types triggered the development of hPSC-derived liver organoids. Therefore, these human liver-based platforms hold great potential for clinical applications. In this review, the production of the different hepatic cell lineages from hPSCs, including hepatocytes, as well as the emerging strategies to generate hPSC-derived liver organoids will be assessed, while current biomedical applications will be highlighted.

## 1. Introduction

From all the internal organs that constitute the human body, the liver is the largest one, and its endocrine and exocrine properties make it also the largest gland. As an endocrine gland, the liver is responsible for the secretion of several hormones, while the bile constitutes the major exocrine secretion. The liver is therefore a central organ in our body that is responsible for homeostasis throughout the human lifespan, performing a complex array of functions [1,2]. Such functions include glycogen storage, drug detoxification, control of metabolism, regulation of cholesterol synthesis and transport, urea metabolism, immunological activity, and secretion of plasma proteins like albumin [1]. The liver is such an essential player in homeostasis, that liver disease due to genetic or environmental factors, such as hepatitis, fibrosis, cirrhosis, and hepatocellular carcinoma, often results in morbidity and mortality [3]. Actually, liver disease is one of the leading causes of death worldwide and it is estimated that approximately 2 million people die per year, representing 3.5% of global deaths. From the 2 million, 1.16 million deaths are caused by cirrhosis (11th cause of death worldwide) and 0.79 million deaths are caused by hepatocellular carcinoma (16th cause of death worldwide) [4]. In addition to mortality rates, liver disease is estimated to have an impact on over 600 million people around the world [5]. Besides that, for end-stage liver failure and other disorders, orthotopic liver transplantation is the only possible solution, making the liver the second most common solid organ transplantation. Still, transplantation needs are poorly met [4].

Unfortunately, the discovery of novel therapeutics for liver disease is still a major challenge, as in vitro modeling of the in vivo physiological functions of the liver is still not accurate. In vitro models are traditionally based on hepatocyte cultures since they are the major parenchymal cell type, accounting for 60% of the total cells in the organ (80% of the volume), mediating almost all liver functions and being its functional metabolic unit [2]. The gold standard source of hepatocytes for scientific investigation has been freshly isolated primary human hepatocytes (PHHs). However, when in culture these cells lose their ability to proliferate and most of their functions are impaired. Additionally, they have limited supply and present batch-to-batch variability, which greatly limit their potential for clinical applications [6,7]. Other hepatocyte sources usually rely on immortalized or cancer cell lines, as well as fetal liver progenitors or adult liver stem cells. Nevertheless, these sources also present major drawbacks, like the fact that their metabolic enzymes do not entirely resemble the ones in adult hepatocytes, in addition to poor cell survival, proliferation, and availability [6,7].

Accordingly, the generation of hepatocytes derived from human pluripotent stem cells (hPSCs) represents a promising cell source to transform our understanding of liver disease and to change the way it is treated [6]. However, their remarkable potential can only be translated into practice if the capability to direct the differentiation of hPSCs into the different hepatic lineages can be achieved. In fact, considerable progress has been made in the past few years in the development of defined protocols for hepatocyte differentiation, which will be further addressed in detail in this review. Hepatocytes may be the largest cellular component of the liver, but its complexity goes far beyond that. Along with hepatocytes, other cell types account for 40% of the cells in the liver, including cholangiocytes (biliary epithelial cells) and other non-parenchymal cells, like liver sinusoidal endothelial cells (LSECs), Kupffer cells (resident liver macrophages), pit cells (natural killer cells), and hepatic stellate cells (HSCs) [1]. To prove this complexity and heterogeneity within the human liver, recent studies were able to create a human liver cell atlas [8]. However, the challenge is now to determine how these cells arrange themselves to form the three-dimensional (3D) architecture that is so crucial for hepatic function. The growing knowledge related to hPSC differentiation triggered the development of new 3D culture technologies like organoids [9]. Consequently, the first steps for human liver organoid production are now being made. In this review, the production of the different hepatic cell lineages from hPSCs, as well as the emerging strategies to generate hPSC-derived liver organoids will be assessed, highlighting their great potential for biomedical applications in regenerative medicine, disease modelling, drug discovery, and hepatotoxicity.

## 2. Hepatogenesis: Origin and Fates of Hepatic Cells

Within the early embryo, as cells develop, they become progressively restricted in their developmental potency. When the pluripotent state is reached, a PSC has the capacity to produce all cell types from the tissues that constitute the human body [10]. However, much of the knowledge in this area is based on mouse developmental processes. During embryogenesis, more precisely during the process of gastrulation, there is the formation of a transient structure in the region of the epiblast, the primitive streak (PS). The developmental path of the liver has its origin prior to, or shortly after, the beginning of gastrulation. Throughout this process, uncommitted cells migrate through the PS and undergo an epithelial-to-mesenchymal transition (EMT), ending in the generation of the three germ layers: ectoderm, mesoderm, and endoderm [11,12]. Thus, the formation of endoderm, the germ layer that eventually gives rise to the liver, results from the intercalation of the definite endoderm that migrated through the PS and minimally by the pre-existing visceral endoderm [13,14]. Molecularly, the TGFβ family member Nodal is responsible for mammalian endoderm specification [15], with high levels of Nodal promoting endoderm and low levels promoting mesoderm specification [16]. This signaling gradient activates the expression of key transcription factors like EOMES (in high levels) and T (in low levels), leading to the cooperation between EOMES with NODAL-SMAD2/3 signaling to induce the expression of endodermal markers such as CER1, FOXA2, and SOX17 [17].

After gastrulation, endoderm undergoes a series of morphogenetic movements resulting in the formation of the gut tube that, in mammals, is patterned along the anterior–posterior axis into three regions, comprising the foregut, midgut, and hindgut [18]. This patterning results from the action of Wnts, bone morphogenetic proteins (BMPs), and fibroblast growth factors (FGFs), that in a gradient from low levels in the anterior region to high levels in the posterior region, results in an anterior foregut, posterior foregut, and midgut-hindgut fate, respectively [19]. The posterior foregut endoderm contains progenitor cells that can give rise to pancreas, liver, and gallbladder. However, for hepatic specification, the convergence of the cardiac mesoderm [20] and septum transversum mesenchyme (STM) [21] is required. FGFs, like FGF1 and FGF2, from the cardiac mesoderm [22,23], as well as BMPs, like BMP4 and BMP2, from the STM are key players in this process [21]. Additionally, endothelial cells also play a relevant role in hepatic specification [24]. Thus, hepatic endoderm thickens to form a liver diverticulum initiating a budding process into the STM. Soon, hepatic endoderm transit from a columnar to a pseudostratified epithelium constituted by hepatoblasts that proliferate and migrate through the STM to form the liver bud [25]. The transcription factor HHEX controls the transition from columnar to pseudostratified epithelium [25], whereas TBX3 and PROX1 promote hepatoblast proliferation and migration [26,27]. HNF4a is equally essential for further development of the liver bud structure [28]. Hepatoblasts are bipotent, having the potential to differentiate into either hepatocytes or cholangiocytes in a process regulated by transforming growth factor-beta (TGFβ), Notch, Wnt, BMP, and FGF [19,29]. Cells committed to a hepatocyte fate progressively mature, and together with cholangiocytes and non-parenchymal cell types, give rise to the final structure of the adult liver (for extend review on liver development see [1,30]).

## 3. Differentiation of Hepatic Cell Lineages from Human Pluripotent Stem Cells

Since the isolation of embryonic stem cells (ESCs) from the blastocyst [31,32,33], and later with the recapitulation of pluripotency in vitro by reprogramming somatic cells into induced pluripotent stem cells (iPSCs) [34,35,36], numerous protocols have been described for the direct differentiation of PSCs into many cell populations across the three germ layers, including neurons [37], cardiomyocytes [38], and also hepatic cells (Table 1). Most of the research in differentiating hepatic cell types from hPSCs has been focused on hepatocytes, but differentiation strategies that originate other enriched populations of liver cells have also been developed. The design of such protocols should bring a new level of complexity to the study of liver development and medical research.

The majority of the differentiation protocols already published are at their core a recapitulation of the natural developmental processes that occur during embryogenesis. Thus, the differentiation of PSCs into a certain cell population can be achieved through the addition of extrinsic signals that guide the cells into a particular fate while repressing alternative ones. These extrinsic signals are usually presented as a combination of signaling pathway agonists and antagonists, added to the culture medium in a stepwise fashion in specific concentrations, sequence, and time [39]. To attain this, it is necessary to understand not only the signaling pathway kinetics, but also the developmental roadmap of the different cell types of the liver, and studying how different cell populations are segregated during fate transitions [40].

### 3.1. Hepatocytes

Since hepatocytes constitute the major cell type in the liver, there has been great interest in differentiating PSCs into this type of cell. The first reports date back to 1996 and relied on the spontaneous differentiation of mouse ESCs using embryoid bodies (EBs) [41]. These EBs spontaneously recapitulate early steps of embryogenesis in an uncontrolled fashion, ending in a highly variable structure with differentiated cells of all three germ layers, where hepatocytes can be present. Only in the following decade the first direct differentiation of hPSCs was reported using hESCs [42] and latter using hiPSCs [43]. Since then, based on developmental signaling pathways and morphogens, many protocols have used similar growth factors and small molecules to generate hepatocytes in adherent culture conditions, leading to outstanding improvements in efficiency and functionality (Table 1).

As previously described, Nodal/Activin, members of the TGF-β superfamily of signaling molecules, are critical for definitive endoderm induction in mammals, and the same is true for in vitro differentiation of PSCs. Kubo and colleagues were the first to provide evidence that high concentrations of Activin A result in endoderm induction [44]. Since then, this growth factor has been widely used in the first steps of hepatocyte differentiation, sometimes associated with other signaling molecules that also promote endoderm development, like BMPs, FGFs, or WNT3A, among others. Afterwards, to recapitulate hepatic specification, BMPs and FGFs have been used to give rise to hepatic endoderm, FGF2, FGF4, BMP4, and BMP2 being the most commonly used. Finally, to trigger hepatoblast proliferation and subsequent hepatocyte differentiation/maturation, hepatocyte growth factor (HGF), Oncostatin M (OSM), and Dexamethasone (DEX) are among the most frequent choices. These differentiation strategies result in the production of what can be called hepatocyte-like cells (HLCs), since current protocols still do not generate hepatocytes with fully mature phenotype when compared to adult hepatocytes. To simplify, in this review we will only use the term “hepatocyte”. Nevertheless, several approaches have been used for in vitro maturation of hPSC-derived hepatocytes, including growth factors, small molecules, transcription factors, and microRNAs [45]. Examples of that are the already mentioned HGF and OSM. It is established that HGF can promote hepatoblast proliferation, migration, and survival, and in the presence of DEX it can upregulate several mature hepatocyte markers [46]. On the other hand, it has also been proven that fetal hepatocytes in the presence of OSM acquire a similar morphology to mature hepatocytes, also contributing to hepatic functionality [46,47].

The majority of the differentiation methods to generate hepatocytes can then be divided into three main stages: endoderm induction, hepatic specification, and hepatocyte differentiation/maturation. However, these three-step approaches may not precisely mimic in vivo liver development, generating impure populations. Trying to overcome this challenge, recent studies attempt to recapitulate hepatocyte development through a sequence of six consecutive lineage choices [40]. On the other hand, there are groups trying to overcome the lack of definition and reproducibility between protocols, relying for that purpose on methods solely driven by small molecules, following a growth-factor-free strategy [48]. Additionally, some studies have been focused on the generation of scalable protocols for large-scale production of hPSC-derived hepatocytes [49,50,51].

### 3.2. Cholangiocytes

Apart from hepatocytes, cholangiocytes have a key role in the liver as they constitute the epithelium lining of the biliary tree, which processes bile production. Some of the first differentiation protocols of hPSCs into hepatocytes reported the presence of bile-duct structures formed by CK7^+^ and CK19^+^ cells, which are well known cholangiocyte cell markers [52,53]. However, the first defined protocol for cholangiocyte differentiation from hPSCs was only published several years later [54]. As mentioned before, during hepatic specification when cells become hepatoblasts, they can give rise to both hepatocytes and cholangiocytes. Therefore, existing protocols for cholangiocyte differentiation follow a similar strategy to obtain hepatoblasts. Once this stage is reached, different strategies have been applied for cholangiocyte specification, including the use of epidermal growth factor (EGF) [54,55], FGF10 [56], TGF-β, and activation of NOTCH signaling (either by using Jagged1 or by the incorporation of OP9 stromal cells in a co-culture system) [55,57].

### 3.3. Other Non-Parenchymal Cells

LSECs are also essential cells of the liver. They are highly specialized endothelial cells that constitute the permeable interface that controls the trafficking of molecules and cells between hepatocytes and the blood stream. Additionally, LSECs play an important role in immunity, liver disease, and regeneration [58,59]. Although important in the maintenance of normal liver functions, there are still few protocols for differentiation of hPSCs into LSECs. In fact, to our best knowledge, there is only one protocol using hPSCs that specifically generates LSECs [60], and a second one using mouse ESCs [61]. Both studies assume that LSECs diverge at some point in ontogeny from endothelial progenitors. Therefore, Koui and colleagues used FLK1^+^CD31^+^CD34^+^ endothelial cells derived from hiPSCs as the starting point to generate and promote a LSEC mature phenotype by adding A83-01 (a TGFβRI inhibitor) under hypoxic conditions. This protocol was able to produce LSECs expressing specific markers like FCGR2B, STAB2, F8, and LYVE1. Generically, many research groups have focused their work on the differentiation of hPSCs into endothelial cells, thus indirectly contributing to future methods for LSEC derivation. For general reviews on this topic, please see [62,63].

Kupffer cells are the liver macrophages, the first line of defense against bacteria, microbial debris, and endotoxins with gastrointestinal origin [64]. In addition to their immunological role, they also cooperate with LSECs in blood clearance [59]. Recently, one study was able to differentiate hiPSCs into Kupffer cells, recapitulating their ontogeny by firstly differentiating hiPSCs into macrophage-precursors and subsequent exposure to hepatic cues by simply culturing these precursors in hepatocyte culture medium (HCM) and advanced DMEM [65].

Finally, HSCs are specialized pericytes that reside in the perisinusoidal space (or space of Disse), between hepatocytes and LSECs. They are important in the maintenance of extracellular matrix homeostasis and its most distinctive feature is the accumulation of vitamin A. Additionally, they play a major role in liver fibrosis upon HSCs activation, i.e., HSCs transdifferentiate from quiescent, vitamin A storing cells into proliferative, fibrogenic myofibroblasts [66,67]. The embryonic origin of HSCs has been and still is elusive, creating a debate around whether these cells originate from the mesoderm, endoderm, or even the neural crest [68]. However, the hypothesis that HSCs originate from the mesoderm (more precisely from the STM) has gained acceptance [69]. This has prompted two studies [60,70] in which hPSCs are firstly differentiated into mesodermal cells, particularly ALCAM^+^ cell populations, and then divergent paths are used to generate HSCs. Koui and colleagues relied solely on the inhibition of the Rho signaling pathway using Y27632 to achieve an HSC phenotype, whereas Coll and colleagues adopted a more complex strategy adding FGF1, FGF3, palmitic acid, and retinol to the culture. This last study claims that not only cells with phenotypic and functional characteristics of mature HSCs are produced, but also that the method is highly robust with around 78% of PDGFRβ^+^ cells and 80% vitamin A-storing cells, a feature of mature HSCs.

## 4. Production of Liver Organoids from Human Pluripotent Stem Cells

The development of advanced 3D culture systems, triggered by the growing knowledge on hPSC differentiation, unlocks the possibility of bringing together multiple organ-specific cell types into a single structure, the so-called organoid [76,77]. Currently, the concept of organoid describes a stem cell-derived 3D structure that through a self-organization process can recapitulate biological features like spatial arrangement, cell–cell, and cell–ECM interactions [9,78]. Accordingly, these structures can provide a much more reliable model of the in vivo anatomy and physiology of a given organ, not only when compared to a 3D cell aggregate composed of a single cell type, but even more when compared to a 2D monolayer culture system. Organoids are thus an emergent system that may serve as building block for tissue engineering applications. The major step forward in this field was given by the groups of Yoshiki Sasai and Hans Clevers in their studies focusing on optic cup [79] and intestinal organoids [80], respectively. Today, scientists can generate organoids from a variety of different cell sources, but some design principles must be taken into consideration when generating an organoid: (1) biophysical properties—the use of solid ECMs to entrap the organoids, or the use of a scaffold-free approach; (2) self-governing of organoid formation—exclusive dependence on endogenous signals or dependence on the addition of exogenous cues; (3) starting cell population—derivation from a single cell, from a homogeneous cell population, or from a co-culture of different cell types [10]. Besides these, recent engineering approaches have been developed to increase the complexity of human organoids and PSCs play an important role in this matter [10,81].

This accumulated knowledge enabled a few groups to recently report the generation of liver organoids derived from hPSCs (Figure 1). The first report of liver organoids dates from 2001 by Michalopoulos and colleagues using several types of adult rat hepatic cells [82], but a more robust and long-term culture of human liver organoids was described later in 2013, using a progenitor population of adult mouse liver [83]. In the same year, Takebe and colleagues were pioneers in using hepatic cells derived from hPSCs, reporting the generation of human liver-like organoids that resemble the developing liver bud during early embryogenesis [84]. These liver bud organoids were generated by mixing hepatic endodermal cells derived from hiPSCs, human umbilical vein endothelial cells (HUVECS) and human mesenchymal stem cells (MSCs). This approach recapitulated the early steps of liver organogenesis resulting in a vascularized human liver bud organoid with improved functionality by producing key liver enzymes. For the maturation of these liver bud organoids, the culture medium used was constituted by HCM and endothelial growth medium (EGM) in 1:1 proportion with the addition of HGF, OSM, and DEX. More recently, the same authors were able to generate fully hPSC-derived liver buds. They used hiPSCs as the cell source to generate hepatic endoderm cells, endothelial cells, and STM cells, and mixed them in a 10:7:2 ratio [85] (Figure 1A). To support the relevance of this approach, complementary in vitro studies demonstrated that besides homotypic interactions between human hepatocytes, heterotypic interactions between hepatocytes and other non-parenchymal cells are critical for self-organization, and that paracrine signals secreted by these cells are important for hepatic maturation [86,87], an idea previously explored in developmental studies using mouse embryos [24]. A similar system was described in 2019 by Pettinato and colleagues using co-cultures of hPSCs with human adipose microvascular endothelial cells (HAMECs) in a 3:1 ratio, that were then submitted to hepatocyte differentiation [88]. This protocol resulted in liver organoids with 89% Albumin^+^ and 15% CD31^+^ cells and improved human hepatic functions associated to mature liver cells. Interestingly, HAMECs self-organized in rosette-like structures within the organoids (Figure 1B). Another strategy used 3D aggregates of hepatoblast-like cells derived from hPSCs that were then co-cultured with human fetal liver mesenchymal cells (hFLMCs) [89]. By day 14 of differentiation, Albumin^+^ cells were found in the peripheral region, whereas PDGFR-β^+^ hFLMCs were found in the center of the organoids. Overall, co-culture of hepatoblast-like cells with hFLMCs, in a 2:1 ratio, generated organoids with increased levels of hepatic functions (Figure 1C).

Apart from co-culture of different cell types as a starting point for the generation of liver organoids, other studies have shown how to start with homogeneous cell populations to obtain complex organoids. Since 2017, two different protocols were published using this approach to produce hPSC-derived liver organoids constituted by hepatocytes and cholangiocytes [90,91]. Both studies started with differentiation of hPSCs into hepatoblasts, but they diverge not only in the approach to get to that point, but also in the way they generate liver organoids. Guan and colleagues started with the production of hepatoblast aggregates that with the addition of exogenous growth factors and subsequent dissociation/reaggregation in Matrigel generated liver organoids with both hepatocytes and cholangiocytes (Figure 1D). On the other hand, Wu and colleagues developed a protocol capable of generating liver organoids with 60% ALB^+^ hepatocytes and about 30% CK19^+^ cholangiocytes. They claimed that the key factors for the success of their study were the inclusion of 25% mTeSR medium during differentiation and the addition of a cholesterol mixture for organoid functional maturation (Figure 1E). Based on a similar approach, a recent work describes the production of hPSC-derived liver organoids containing hepatocytes, HSCs, Kupffer cells, and cholangiocytes [92]. They initially differentiated hPSCs to foregut, collecting foregut spheroids released from the 2D culture and embedding them in Matrigel with further addition of retinoic acid, a molecule that reportedly plays and important role in the specification not only of parenchymal, but also non-parenchymal liver cells (Figure 1F). Apart from these examples using hPSC-derived cells, other human cell types have been used in the generation of human liver organoids in the past few years [83,93]. Likewise, liver organoids have been produced using mouse [94], rat [95], cat [96], and canine cells [97].

It is important to note that organoid technology is becoming an increasing trend in biomedical research [9]. However, some caution needs to be taken into consideration since by definition, an organoid is a reductionist cell construct that captures the cellular, structural, and physiological complexity of a given organ. Therefore, a clear distinction between 3D spheroids made up of a single cell type and without clear self-organization, and organoids needs to be made. In this section we tried to focus only on reports that fit into this organoid concept.

## 5. Applications of hPSC-Derived Hepatic Cell Lineages and Liver Organoids

As demonstrated above, hPSCs have the capacity to establish human liver models giving researchers the opportunity to design human liver-based platforms for disease modeling, drug discovery, and hepatotoxicity. Furthermore, differentiated hepatic cells and liver organoids derived from hPSCs represent a renewable source for cell-based therapies aiming at the treatment of patients suffering from liver disease (Figure 2).

### 5.1. Regenerative Medicine

The use of hPSC-derived cells for regeneration or replacement of damaged tissue in regenerative medicine has been proposed to deliver functional recoveries [98]. Indeed, hPSC-based therapies were already established in several clinical trials [99]. hiPSCs in particular have the important advantage of their capability in generating differentiated patient-specific cells, allowing autologous cell transplantation and, theoretically, suppressing the risk of immune rejection. This milestone was achieved in 2014 with the successful transplantation of autologous retinal pigment epithelium sheets derived from hiPSCs [100]. Besides all this progress, chromosomal aberrations (due to cell reprograming and subsequent culture), as well as the tumorigenicity of undifferentiated cells, represent some of the hurdles that need to be taken into consideration when using hPSCs in cell therapies [98,101].

As mentioned above, orthotopic liver transplantation is the single solution for end-stage liver failure and other liver disorders, making this organ the second most common solid transplantation after kidney. Still, given the current transplantation rates, less than 10% of global organ transplantation needs are met [4]. This fact results in the need for alternative therapeutic strategies, and hepatocyte transplantation has been perceived as one [102]. The first attempt to use these cells for transplantation dates back to 1976 using Gunn rats as animal models for Crigler–Najjar syndrome [103]. Still, it was only in 1992 that primary hepatocytes were transplanted into human patients [104]. Since then, numerous patients with liver disease have been treated with hepatocyte transplantation. However, this strategy still presents major hurdles like the limited source of hepatocytes, poor quality of isolated cells, and occasional hepatocyte rejection [102]. Therefore, hPSC-derived hepatocytes have been seen as a potential cell source for transplantation (Table 2). In fact, one of the most successful studies using hPSC-derived hepatocytes for transplantation was able to repopulate up to 15% of the liver of uPA immunodeficient mice after intrasplenic injection [105]. In this study, the differentiated hepatocytes still presented fetal markers like AFP and they were largely negative for isoforms of CYP450. However, after transplantation, the transplanted cell population acquired mature features such as downregulation of AFP expression and cells positive for CYP450 isoforms. This transplantation strategy has also been successful in the alleviation of liver metabolic disorders [106] and in acetaminophen-induced acute toxicity [107], with liver repopulation rates of 2.5–7.5% and 10%, respectively. Thus, hPSC-derived hepatocytes demonstrated their potential to become a relevant cell source for liver cell therapy. However, although initial studies are promising, the use of these cells for regenerative medicine applications needs to be more efficient and effective until it can be translated into human benefit. To accomplished that, recent studies have been using different strategies besides intrasplenic injection for hepatocyte transplantation. One of these strategies is the transplantation of hepatocyte sheets onto the surface of mice livers [108]. Additionally, different scaffolds have been used to support hPSC-derived hepatocytes for subsequent transplantation, namely PCL fibers [109] and decellularized livers [110,111]. Besides this, a recent study reported for the first time, to the best of our knowledge, the use of current good manufacturing practice (cGMP)-compliant hepatocytes generated from hPSCs for transplantation [112]. Liver regeneration will probably require more than simply injecting the right type of cells in the right place. The foundation of knowledge concerning liver regeneration mechanisms, both in normal development and after injury, seems to provide a strong platform to achieve this goal.

### 5.2. Disease Modeling

In addition to regenerative medicine, disease modeling constitutes another important biomedical application for hPSC derivatives. In vitro disease models based on hPSC technology should improve our knowledge regarding pathological mechanisms underlying human diseases, either genetic or acquired [115]. Given the relevance of liver disease, several studies using hPSC-derived hepatocytes have been published in the last years (for extended review see [7]). However, the effort for improved maturity and greater complexity of in vitro culture systems for disease modeling has led to recent publications using liver organoids as a platform to study liver disease (Table 3).

With the advent of novel gene editing tools like CRISPR-Cas9, it is now possible to induce disease-causing mutations or silencing mutations carried by patient-specific hPSCs and evaluate their effects in differentiated cell phenotypes [116,117]. To understand liver disease mechanisms at the organ level, at least two different studies have been published studying genetic diseases in hPSC-derived liver organoids. Guan and colleagues used patient-specific hPSCs to model Alagille Syndrome (ALGS) and Tetralogy of Fallotthis (TOF) genetic disorders [90]. Firstly, they generated liver organoids from hPSCs reprogrammed from ALGS patients, where in contrast to healthy organoids, mature hepatocytes were developed, but cholangiocytes and bile ductular structure development was impaired. Additionally, they used CRISPR-Cas9 technology to introduce and revert an ALGS causing *JAG1* mutation, C829X, into control and ALGS hPSCs. Thus, ALGS liver pathology was recapitulated, and it was also shown that *JAG1* haploinsufficiency alone does not produce pathology in liver organoids. Moreover, this team also modelled a disease caused by another mutation in *JAG1*, TOF, demonstrating that the type of *JAG1* mutation has a considerable effect in the onset of liver disease. More recently, another study has demonstrated that liver organoids are a suitable platform to model steatohepatitis, a condition that is, among others, characteristic of Wolman disease, caused by a defective activity of lysosomal acid lipase (LAL) [92]. Firstly, these researchers induced steatohepatitis phenotype in liver organoids exposing them to free fatty acids, resulting in lipid accumulation, inflammation, and fibrosis. After that, to highlight the clinical relevance of modelling steatohepatitis, they used patient-derived hPSCs with LAL deficiency to generate liver organoids, thus recapitulating the Wolman disease phenotype with severe steatohepatitis. Additionally, it was demonstrated through liver organoid technology that the steatohepatitis phenotype could be rescued using FGF19, suppressing lipid accumulation and improving liver organoids survival. Besides these two examples of genetic disease modeling, organoids derived from adult liver tissue were already used to study A1AT deficiency and Alagille syndrome [93].

Recently, liver disease modelling has also been successfully performed to study acquired liver diseases. An example is hepatitis B virus (HBV) infection of hPSC-derived liver organoids [118]. This culture system proved to be more susceptible to HBV when compared to hepatocytes differentiated in a 2D culture system. Particularly, the infection of liver organoids with HBV resulted in hepatic dysfunction with downregulation of hepatic gene expression and emergence of hepatic injury markers, along with the alteration of hepatic structures. Therefore, this study suggested that liver organoids can be considered a good platform for HBV modelling, recapitulating the virus life cycle and consequent dysfunctions. Another example of disease modeling of acquired liver diseases using liver organoids is the study of alcoholic liver disease (ALD), the number one cause of liver-associated mortality in Western countries [89]. Upon EtOH treatment for 7 days, liver organoids displayed liver damage and reduction in cell viability, as well as upregulation of gene expression of fibrogenic markers, thus recapitulating ALD pathophysiology. Additionally, EtOH treatment led to enhanced oxidative stress, an established characteristic of ALD that starts with the metabolism of EtOH by ADH and CYP2E1. Once more, liver organoids proved to be a reliable platform for disease modeling, encouraging its use to study new conditions and eventually contributing to the discovery of new therapeutics.

It is important to note that the cell composition of liver organoids can be of extreme importance when modeling liver diseases. In the examples above, it is possible to understand that given the biliary deficiencies in ALGS and TOF, the presence of cholangiocytes within these organoids it is an essential requirement [90]; similarly, given the characteristic fibrosis of steatohepatitis, HSCs should also be present [92]. Obviously, increasing the complexity of the model system will result in better recreating liver function, and it may even expose the role of the different hepatic cellular components in disease development. In fact, a very recent study shows how the crosstalk between hepatocytes, hepatic Kupffer cells, and HSCs play an important role in alcoholic liver disease (ALD), providing new insights into this pathology and identifying potential new targets for drug therapy [119,120].

### 5.3. Drug Discovery and Hepatotoxicity

Modeling of human diseases is driven by the need for novel therapeutics aiming at disease treatments and cures. For this reason, drug discovery and toxicological assays are considered a potential application for hPSC derivatives [115,121]. To this end, animal models have been continuously used for drug screening. However, differences between the actual human setting and other animals result in inaccurate prediction of drug effects. Moreover, animal models are not suitable for high-throughput screening of small-molecule libraries [116,122]. As an alternative, the use of hPSC-based models for drug screens have been amply established, assessing not only the efficacy of potential drug candidates, but also their toxicity, predicting the likelihood of potential drugs to cause severe side effects [98]. It is also important to bear in mind that each patient has a specific genetic background, and that this fact implies different responses to medication. Accordingly, hepatocytes and liver organoids generated from hPSCs can be used as a new tool to investigate not only disease mechanisms, but also therapeutic strategies, creating the foundation for personalized therapies, an emerging approach known as precision medicine [122]. Currently, pharmaceutical development is highly costly ($2.6 billion per drug that enters the market) and inefficient (89% of drugs that enter clinical trials will fail due to unforeseen toxicity) [7,123]. Part of this is due to inadequate screening during preclinical studies and so, the use of hPSC-derived hepatocytes and liver organoids in this field is of extreme relevance, as hepatotoxicity is the major type of toxicity associated to drug withdrawals (21% of the cases) [124].

Given this context, there is the urgent need to create protocols that can generate hepatocytes or liver organoids in a scalable and miniaturized fashion, being suitable for high throughput screening of small molecule libraries. Examples of this have been already published [85,125]. These high-throughput screening platforms were already used for identification of drugs for disease treatment with hPSC-derived hepatocytes. At least three studies used small molecules/drug libraries aiming the attenuation or reversion of the effects of diseases like alpha-1-antitrypsin (AAT) deficiency [126], familial hypercholesterolemia [127], and mitochondrial DNA depletion syndrome (MTDPS3) [128]. The same platforms have also been used for toxicity screens evaluating the effect of certain drugs on hPSC-derived hepatocytes, typically testing compounds known to be toxic and non-toxic and assessing cell morphology and viability [129,130,131].

All of these studies are mainly focused on hepatocytes, but as described before non-parenchymal cells hold great importance in liver physiology. For instance, LSECs are implicated in most liver diseases making them an attractive therapeutic target [58]. Additionally, Kupffer cells play a crucial role in drug-induced liver injury (DILI) and other liver diseases. To demonstrate this, hepatocytes have been co-cultured with Kupffer cells resulting in a model that is more sensitive in detecting hepatotoxicity induced by different drugs [65]. Thus, the development of more complex liver organoids composed of different hepatic cell types can substantially benefit drug discovery and hepatotoxicity assays.

Drug screening can also be performed using microfluidic devices like the so-called organ-on-a-chip or microphysiological systems (MPSs), where living cells can be cultured in continuously perfused chambers, modeling the physiological functions of a given tissue or organ [132,133]. Accordingly, organ-on-a-chip is also a valuable platform for drug development and toxicology giving insights into adsorption, distribution, metabolism, elimination, and toxicity (ADMET), mathematical pharmacokinetics (PK), pharmacodynamics (PD), and drug efficacy [134]. In fact, Wang and colleagues have recently achieved the in situ differentiation of hPSCs into liver organoids using a perfusable micropillar chip [135]. The on-chip liver organoids displayed both hepatocytes and cholangiocytes, as well as increased cell viability and mature cell signature. Notably, the organoids generated in this platform not only presented high levels of cytochrome P450 enzyme expression, but also dose- and time-dependent hepatotoxic response to acetaminophen. These results support the notion that organ-on-a-chip technology constitutes a valid platform for drug testing. Moreover, organ-on-a-chip technology can rely not only on individual designs, but also in more complex interlinked multi-organ-on-chips, or body-on-a-chip platforms capable of mimicking multi-organ crosstalk [136]. Indeed, the study of an inter-tissue crosstalk between gut and liver during inflammatory processes was already reported [137], and more recently an interconnected MPS representing up to 10 organs, including liver, was established [138]. Applying this technology to hPSC-derivatives is still, to the best of our knowledge, an unmet need.

## 6. Conclusions

Stem cell research has made considerable progress in the last years, providing researchers with the necessary tools to replicate in vitro the developmental processes from hepatogenesis and, ultimately, the physiological and structural features of human liver. To that end, numerous strategies have been published for hPSC direct differentiation into hepatocytes, cholangiocytes, and other non-parenchymal hepatic cell types. However, some of these cells still present a fetal phenotype with only a certain degree of maturity. Recently, organoid technology has contributed to the increasing complexity of hepatic culture systems in vitro, leading to enhanced metabolic functionality and cellular architecture, and furthering our knowledge of liver biology. As new technologies are being developed, the potential applications of hepatic cell lineages and liver organoids derived from hPSCs are increasing, bringing new insights into the fields of regenerative medicine, disease modelling, drug discovery, and hepatotoxicity. Thus, it is in the intersection of biology and engineering that many open questions concerning the human liver can be brought to light, leading, eventually, to an improvement in the human condition.

## Figures and Tables

**Figure 1 bioengineering-07-00036-f001:**
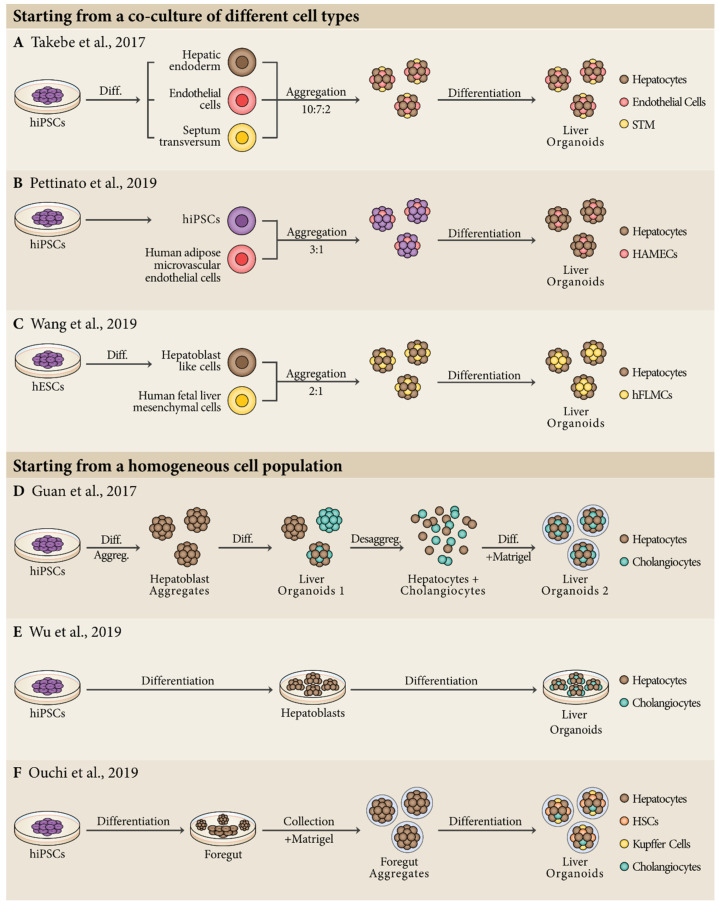
Current strategies for the generation of human pluripotent stem cell (hPSC)-derived liver organoids. So far, liver organoids have been generated by (**A**–**C**) co-culture of different cell types, including hPSCs, differentiated hepatic cell lineages, or isolated human cells with potential to promote liver organoid differentiation/maturation; (**D**–**F**) homogeneous cell populations that, through differentiation, are capable of generating cellular constructs with structural and physiological complexity. STM, septum transversum mesenchymal cells; HSCs, hepatic stellate cells.

**Figure 2 bioengineering-07-00036-f002:**
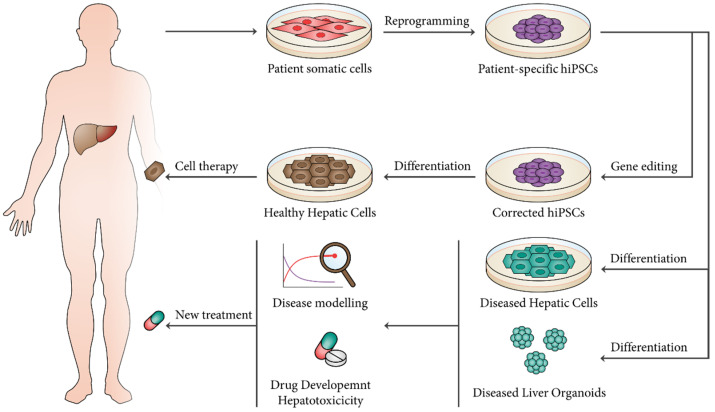
Clinical applications for hPSC-derived hepatic cells and liver organoids. Isolated somatic cells from patients can be cultured and reprogrammed into patient-specific hiPSCs. These cells represent a promising cell source for cell therapy, as differentiated hepatocytes can be used for transplantation in regenerative medicine strategies. Additionally, differentiated hepatic cell lineages or generated liver organoids can be applied in disease modelling, as well as drug development and hepatotoxicity assays.

**Table 1 bioengineering-07-00036-t001:** Methods for differentiation of human pluripotent stem cells into hepatic lineages.

Study	Media	Molecules	Ref.
**Hepatocytes**			
Rambhatla et al., 2003	KO-DMEM+FBS	NaB/DMSO – NaB/HGF	[42]
Kubo et al., 2004	StemPro34 – IMDM+SR	Act A – DEX	[44]
Hay et al., 2008	RPMI+B27 – DMEM+SR+DMSO – L15	Act A/Wnt3a – HGF/OSM	[71]
Song et al., 2009	RPMI – HCM – N2B27	Act A – FGF4/BMP2 – HGF/KGF – OSM/DEX	[43]
Si-Tayeb et al., 2010	RPMI+B27 – HCM	Act A – BMP4/FGF2 – HGF – OSM	[72]
Sullivan et al., 2010	RPMI+B27 – DMEM+SR+DMSO – L15	Act A/Wnt3a – Act A – HGF/OSM	[73]
Touboul et al., 2010	CDM	Act A/Ly/BMP4/FGF2 – FGF10 – FGF10/RA/SB – FGF4/HGF/EGF	[74]
Kajiwara et al., 2012	RPMI+B27 – DMEM+SR+DMSO – HCM	Act A/Wnt3a/NaB – Act A/Wnt3a – HGF/OSM	[75]
Siller et al., 2015	RPMI+B27 – DMEM+SR+DMSO – L15	CHIR – Dihexa/DEX	[48]
Ang et al., 2018	CDM	Act A/CHIR/PI – Act A/LDN – A83/BMP4/FGF2/ATRA – Act A/CHIR/BMP4/Forskolin – BMP4/OSM/DEX/Forsk/Ro/AA/Insulin – DEX/Forskolin/Ro/AA/Insulin	[40]
**Cholangiocytes**			
Dianat et al., 2014	RPMI	Act A/Ly – Act A/FGF2/BMP4 – FGF4/HGF/EGF/RA – EGF/GH/IL6	[54]
De Assuncao et al., 2015	RPMI – H69	Act A/Wnt3a – FGF2/BMP4/SHH – SHH/JAG1 – TGFβ	[57]
Ogawa et al., 2015	RPMI – H16 – H16/Ham’s F12 – H21/Ham’s F12	Act A/CHIR – FGF2/BMP4 – HGF/OSM/DEX – HGF/EGF/TGFβ/OP9	[55]
Sampaziotis et al., 2015	CDM – RPMI – William’s E	Act A/Ly/FGF2/BMP4 – Act A – BMP4/SB – Act A/FGF10/RA	[56]
**Liver Sinusoidal Endothelial Cells**			
Koui et al., 2017	StemPro34 SFM – EGM2	BMP4 – BMP4/Act A/FGF2 – VEGF/SB/Dorsomorphin – VEGF – A83	[60]
**Kupffer Cells**			
Tasnim et al., 2019	mTeSR1 – X-VIVO – PHCM/Adv DMEM	BMP4/VEGF/SCF – MCSF/IL3 – MCSF	[65]
**Hepatic Stellate Cells**			
Koui et al., 2017	StemPro34 SFM – MSCGM	BMP4 – BMP4/Act A/FGF2 – VEGF/SB/Dorso – ROCKi	[60]
Coll et al., 2018	MCDB 201	BMP4 – BMP4/FGF1/FGF3 – FGF1/FGF3/PA/Retinol	[70]

The symbol “/” is used to separate media and molecules within the same differentiation step and the symbol “–“ is used to separate the different differentiation steps. NaB, sodium butyrate; Act A, Activin A; Ly, Ly294002; SB, SB431542; CHIR, CHIR99021; PI, PI103; LDN, LDN193189; A83, A8301; Ro, Ro4929097; ROCKi, Y-27632.

**Table 2 bioengineering-07-00036-t002:** Summary of recent studies on hPSC-derived hepatocyte transplantation in murine models.

Study	Route	Cells	Nr of Cells	% Repopulation	Ref.
Carpentieret al., 2014	Intrasplenic injection	hPSC-hepatocytes	4 × 10^6^	<1–20%	[105]
Chen et al., 2015	Intrasplenic injection	hPSC-hepatocytes	2 × 10^6^	2.5–7.5%	[106]
Tolosa et al., 2015	Intrasplenic injection	hPSC-hepatocytes	1 × 10^6^	10%	[107]
Nagamoto et al., 2016	Sheet transplantation	hPSC-hepatocyte sheet	8 × 10^5^	-	[108]
Takayama et al., 2017	Intraperitoneal transplantation	hPSC-hepatocytes	1 × 10^6^	-	[113]
Nie et al., 2018	Renal subcapsular space	hPSC-hepatocyte aggregates	1 × 10^6^	-	[114]
Rashidi et al., 2018	Intraperitoneal transplantation	hPSC-hepatocyte aggregates	2 × 10^6^ (aggregates)	-	[109]
	Subcutaneous transplantation	hPSC-hepatocytes in PCL fibers	-	-	[109]
Blackford et al., 2019	Intraperitoneal transplantation	hPSC-hepatocyte aggregates	2 × 10^3^ (aggregates)	-	[112]

**Table 3 bioengineering-07-00036-t003:** Reported studies for disease modeling in hPSC-derived liver organoids.

Study	Disease	Gene/Toxin	Approach	Ref.
**Genetic Liver Diseases**				
Guan et al., 2017	Alagille syndrome	JAG1	Patient-derived/gene editing	[90]
	Tetralogy of Fallot	JAG1	Patient-derived	[90]
Ouchi et al., 2019	Wolman disease	Free fatty acids	Patient-derived/induced	[92]
**Acquired Liver Diseases**				
Nie et al., 2018	Hepatitis B	Hepatitis B virus	Induced	[118]
Wang et al., 2019	Alcoholic liver disease	EtOH	Induced	[89]

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
