# Peer review of "Production of Human Pluripotent Stem Cell-Derived Hepatic Cell Lineages and Liver Organoids: Current Status and Potential Applications"

_bioengineering, 2020, doi:10.3390/bioengineering7020036_

Round 1
Reviewer 1 Report
Liver disease is major cause of death around the world. Orthotopic liver transplantation is required to treat liver disease, but the donor organ shortage is a serious problem. Additionally, there are no appropriate in vitro models that mimic in vivo liver physiology. Here, in this review, Cotovio et al. focused on hepatic cell lineages derived from human pluripotent stem cells (hPSCs) as a cell source for liver cell therapy, disease modeling, and drug discovery. Although it is well written and summarized in this area, some improvements are necessary. Please address the following questions and comments.
- The authors said “it was only in 1992 that autologous hepatocytes were transplanted into human patients.”(line 330) Is it correct? I assume that patients were transplanted with allogeneic hepatocytes.
- It is described that “the use of these cells for regenerative medicine applications needs to be more efficient and effective.” (line 343) There are various methods for transplantation to deliver hepatocytes, such as intrasplenic injection, sheet transplantation, PCL fibers, or decellularized liver scaffolds. Therefore, the authors could summarize such studies. The following papers are examples.
Nagamoto et al., J Hepatol., 2016
Lorvellec et al., PLoS One., 2017
Takayama et al., Hepatol Commun., 2017
Rashidi et al., Arch Toxicol., 2018
Nie et al., Stem Cell Res Ther., 2018
Minami et al., Regen Ther., 2019
Author Response
Response to Reviewer 1 Comments
Point 1: The authors said, “it was only in 1992 that autologous hepatocytes were transplanted into human patients.” (line 330) Is it correct? I assume that patients were transplanted with allogeneic hepatocytes.
Response 1:
We thank the reviewer for bringing this point to our attention. In fact, the aforementioned study does include autologous hepatocyte transplantation as can be seen in the article: “A 52-year-old male with chronic hepatitis and cholelithiasis accepted autotransplantation of a total of 1 x 108 hepatocytes harvested from his lateral segment during cholecystectomy.”. However, as the important point is not whether autologous or allogeneic transplantation was done, but the fact that it is the first transplantation of adult hepatocytes in humans, the term autologous was replaced by primary (line 330).
Point 2: It is described that “the use of these cells for regenerative medicine applications needs to be more efficient and effective.” (line 343) There are various methods for transplantation to deliver hepatocytes, such as intrasplenic injection, sheet transplantation, PCL fibers, or decellularized liver scaffolds. Therefore, the authors could summarize such studies. The following papers are examples. Nagamoto et al., J Hepatol., 2016; Lorvellec et al., PLoS One., 2017; Takayama et al., Hepatol Commun., 2017; Rashidi et al., Arch Toxicol., 2018; Nie et al., Stem Cell Res Ther., 2018; Minami et al., Regen Ther., 2019.
Response 2:
We acknowledge the relevance of the point raised by the reviewer. Therefore, to address the several methods for hepatocyte transplantation the following additions were made to the manuscript:
- “…(Table 2).” – line 334.
- “…after intrasplenic injection – line 336.
- “To accomplished that, recent studies have been using different strategies besides intrasplenic injection for hepatocyte transplantation. One of these strategies is the transplantation of hepatocyte sheets onto the surface of mice livers [110]. Additionally, different scaffolds have been used to support hPSC-derived hepatocytes for subsequent transplantation, namely PCL fibers [111] and decellularized livers [112,113].” – line 345 to 349.
- Table 2. Summary of recent studies on hPSC-derived hepatocyte transplantation in murine models. – line 355.
Study |
Route |
Cells |
Nr of cells |
% Repopulation |
Ref. |
Carpentieret al., 2014 |
Intrasplenic injection |
hPSC-hepatocytes |
4x106 |
<1%-20% |
[107] |
Chen et al., 2015 |
Intrasplenic injection |
hPSC-hepatocytes |
2x106 |
2.5%–7.5% |
[108] |
Tolosa et al., 2015 |
Intrasplenic injection |
hPSC-hepatocytes |
1x106 |
10% |
[109] |
Nagamoto et al., 2016 |
Sheet transplantation |
hPSC-hepatocyte sheet |
8x105 |
- |
[110] |
Takayama et al., 2017 |
Intraperitoneal transplantation |
hPSC-hepatocytes |
1x106 |
- |
[115] |
Nie et al., 2018 |
Renal subcapsular space |
hPSC-hepatocyte aggregates |
1x106 |
- |
[116] |
Rashidi et al., 2018 |
Intraperitoneal transplantation |
hPSC-hepatocyte aggregates |
2x106 (aggregates) |
- |
[111] |
Subcutaneous transplantation |
hPSC-hepatocytes in PCL fibers |
- |
- |
[111] |
Blackford et al., 2019 |
Intraperitoneal transplantation |
hPSC-hepatocyte aggregates |
2x103 (aggregates) |
- |
[114] |
Also, wherever the former Table 2 was mentioned, it was now changed to Table 3 (line 365 and 417).

Reviewer 2 Report
The authors Cotovio and Fernandes discuss in their manuscript the production of all almost all hepatic cell lineages derived from human pluripotent stem cells and adumbrate the production of liver organoids from these cell type and possible applications in different .
The review is very elegant and easy to follow, the language used here is very perspicuous. The figures are very plausible.
The manuscript exhibits the great knowledge of the authors in the field of pluripotent stem cell engineering as they have already shown as authors and editor of the book Engineering Strategies for Regenerative Medicine (DOI 10.1016/B978-0-12-816221-7.00001-X edited by Fernandes, Diogo, Cabral).
Minor point:
Since the authors also address the clinical application, the readers could be interested in the current status of GMP of the human pluripotent stem cells, please could authors consider to discuss that part.
Author Response
Response to Reviewer 2 Comments
Point 1: Since the authors also address the clinical application, the readers could be interested in the current status of GMP of the human pluripotent stem cells, please could authors consider discussing that part.
Response 1:
We would like to thank the reviewer for this pertinent question. In addition to our response to reviewer 1, in which several methodologies for hepatocyte transplantation were added to the manuscript, a reference to cGMP-compliant hepatocytes generated from hPSCs was also added to the main text and to the new Table 2:
“Besides this, a recent study reported for the first time, to the best of our knowledge, the use of current good manufacturing practice (cGMP)-compliant hepatocytes generated from hPSCs for transplantation [114].” (line 349-351).
